# Mothers’ Breast Milk Composition and Their Respective Infant’s Gut Microbiota Differ between Five Distinct Rural and Urban Regions in Vietnam

**DOI:** 10.3390/nu15224802

**Published:** 2023-11-16

**Authors:** Guus A. M. Kortman, Harro M. Timmerman, Anne Schaafsma, Eline Stoutjesdijk, Frits A. J. Muskiet, Nguyen V. Nhien, Els van Hoffen, Jos Boekhorst, Arjen Nauta

**Affiliations:** 1NIZO Food Research B.V., 6718 ZB Ede, The Netherlands; 2Friesland Campina, Stationsplein 4, 3818 LE Amersfoort, The Netherlands; 3Department of Laboratory Medicine, University Medical Center Groningen, University of Groningen, 9713 GZ Groningen, The Netherlands; 4National Institute of Food Control, No. 65 Pham Than Duat Str., Mai Dich Ward., Cau Giay Dist., Ha Noi 100000, Vietnam

**Keywords:** breast milk, infant, microbiota, diet, nutrients, oligosaccharides

## Abstract

Microbiota colonization and development in early life is impacted by various host intrinsic (genetic) factors, but also diet, lifestyle, as well as environmental and residential factors upon and after birth. To characterize the impact of maternal nutrition and environmental factors on vaginally born infant gut microbiota composition, we performed an observational study in five distinct geographical areas in Vietnam. Fecal samples of infants (around 39 days old) and fecal and breast milk samples of their mothers (around 28 years) were collected. The microbiota composition of all samples was analyzed by 16S rRNA gene Illumina sequencing and a bioinformatics workflow based on QIIME. In addition, various breast milk components were determined. Strong associations between the geographically determined maternal diet and breast milk composition as well as infant fecal microbiota were revealed. Most notable was the association of urban Ha Noi with relatively high abundances of taxa considered pathobionts, such as *Klebsiella* and *Citrobacter*, at the expense of *Bifidobacterium*. Breast milk composition was most distinct in rural Ha Long Bay, characterized by higher concentrations of, e.g., docosahexaenoic acid (DHA), eicosapentaenoic acid (EPA), selenium, and vitamin B12, while it was characterized by, e.g., iron, zinc, and α-linolenic acid (ALA) in Ha Noi. Breast milk iron levels were positively associated with infant fecal *Klebsiella* and negatively with *Bifidobacterium*, while the EPA and DHA levels were positively associated with *Bifidobacterium*. In conclusion, differences between five regions in Vietnam with respect to both maternal breast milk and infant gut microbiota composition were revealed, most likely in part due to maternal nutrition. Thus, there could be opportunities to beneficially steer infant microbiota development in a more desired (rural instead of urban) direction through the mother’s diet.

## 1. Introduction

It is well established that the human gut microbiota have a significant impact on health and wellbeing due to their bi-directional relationship [1]. In addition to the protective role of the gut microbiota towards invading pathogens, as well as the breakdown of food components and nutrient metabolism, the gut microbiota also contribute to gut physiology, gut mobility, and intestinal barrier homeostasis. In addition, the gut microbiota have a crucial role in the development, training, and functionality of the innate and adaptive immune system and facilitate the communication between the gut and the brain via the so-called gut–brain axis, impacting brain function, behavior, and mental well-being (mood, stress, and anxiety) [2].

The scene for a healthy microbiota composition and activity is set upon birth [1,3]. The subsequent microbiota colonization and development in early life, as impacted by various host intrinsic (genetic) factors, but also diet, life style, as well as environmental and residential factors upon and after birth, are of eminent importance [4,5,6,7]. It has become apparent that microbiota composition seems to be predominantly shaped by non-genetic factors [5]. Thus, there could be opportunities to beneficially impact the early life microbiota development trajectory.

A gradual colonization of the infant gut is stimulated by a term and vaginal birth and breastfeeding, enabling the maternal transmission of beneficial microbiota members and their subsequent stimulation towards a resilient cross-feeding microbiota consortium [8,9]. Breast milk is not only of eminent importance to nourish and protect the newborn infant, but also shapes the infant’s developing gut microbiota. It contains essential micro- and macronutrients for the newborn, and, in addition, provides oligosaccharides, biologically active components, and live bacteria that play critical roles in microbiota development [10]. Although breast milk is the gold standard for young infants, the nutritional, bioactive, and microbiota composition is variable and impacted by maternal genetics and lactation stage, but also by lifestyle and diet [11,12,13,14,15]. For example, different dietary patterns due to geographic region, socioeconomic, and nutritional status have been shown to impact breast milk composition with respect to, e.g., the total amount and relative abundance of human milk oligosaccharides (HMOs), breast milk microbiota, and the concentration of docosahexaenoic acid (DHA) and arachidonic acid (AA) [16,17,18,19,20]. Also, differences in omega-3 and omega-6 long-chain polyunsaturated fatty acids (PUFAs) were revealed between urban and suburban mothers, as ascribed to rural and urban alimentary habits [20]. The respective (both positive and negative) impact of the maternal diet on the breastfed infant gut microbiome has also become apparent on the basis of several studies, as recently reviewed by Taylor et al. (2023) [21].

Notably, factors related to (rural, urbanized, or even westernized) lifestyle and environment have also been shown to impact microbiota development as exemplified by urbanization, with underlying changes in socioeconomic status, diet, environment, antibiotics use, caesarian sections, and hygiene/sanitization practices [22,23,24,25,26,27,28]. Distinctive signatures of the response to urbanization in the microbiota development process were revealed in both adults and infants [22,23,28]. Ayeni et al. (2018) revealed differences in the fecal microbiome and metabolome between an isolated group of rural agriculturalists and urban individuals from capitals, also for infants aged <3 years [22]. Specific microbiota traits such as the dominance of typical oligosaccharide/fiber degraders and the relatively low inter-individual variation between infants were shown to be lost upon urbanization. Their findings support the likely existence of distinct trajectories of development of the intestinal ecosystem in early life, depending on geography and, thus, on cultural and socioeconomical contexts. Notably, breast milk microbiota composition has also been shown to be impacted by urbanization, with urban mothers exhibiting a significant higher abundance of Proteobacteria, including the *Klebsiella* genus [29]. As the gut microbiota assembly process is of importance for health and disease, urban-associated practices may increase the susceptibility to pathobiont outgrowth, and even have adverse effects on (immune and metabolic) health in the long term [23,30].

In this study, we set out to reveal the differences in mothers’ breast milk composition and the gut microbiota composition of their respective vaginally born, mainly breast-fed infants around 39 days after birth in five distinct geographical areas in rural (Tien Giang, Phu Tho, Ha Long Bay) and urban (Ha Noi, Ho Chi Minh City) Vietnam. Vietnam is a diverse country in terms of geography (extends over 1000 miles from north to south), genetic background, and cultural and dietary habits. The five regions were selected on basis of various eminent differences; Tien Giang (rural, inland), Ha Long Bay (rural, coast), and Ha Noi (urban) are in the north, while Phu Tho (rural) and Ho Chi Minh City (urban) are situated in the south. Typical dietary habits also differ between these areas, not only between the rural and urban areas, but also between the different rural areas. We characterized several aspects that could be discriminative and impact infant fecal microbiota composition, i.e., maternal diet, household composition, maternal secretor status, breast milk nutrient/oligosaccharide composition, breast milk microbiota composition, and mother fecal microbiota composition. We revealed strong associations between the geographical region, breast milk composition, and infant’s microbiota composition that could provide opportunities to explore ways to beneficially impact the early life microbiota development trajectory.

## 2. Material and Methods

### 2.1. Study Cohort and Sampling

The study cohort and sampling has been described before [31]. In brief, healthy and well-nourished (self-proclaimed and determined by visual observation) lactating mothers with their healthy infant (vaginally born and predominantly breastfed), having at least one prior child, were invited to participate in the study. The study, “Relationship between Vietnamese breast milk composition, maternal diet and maternal and offspring microbiome”, was approved on 18 July 2013, by the Ethics Committee of The Family Food and Nutrition Institute in Ha Noi, Vietnam. All women gave informed and written consent for their participation and for the collection of data and feces samples of their children. The study complied with the Helsinki declaration of 1975 as revised in 2013. Milk (n = 91) and fecal (n = 82) samples were collected from 100 mothers (average age of 28 years, around 20 per region), and fecal samples (n = 78) from their infants (average age of 39 days). In total, 68 breast milk-infant feces sample pairs were obtained. The milk sample and feces sample were taken at the same time, between 3- and 6-weeks post-partum. The mothers were instructed to save a 25 mL milk sample that was taken from a completely emptied breast around noon. Tubes were immediately frozen and stored at −80 °C. Feces were sampled (5–10 g) from mother and infant in stool collection tubes with a spoon attached to the lid, and stored at cold temperature within 1 h. These samples were transported on ice to the nearest −20 °C freezer as soon as possible. Shipment to the Netherlands took place on dry ice with temperature control. The samples were stored at −20 °C until further analysis. The dietary and health information of infants and mothers was collected via an anthropometry questionnaire, health status questionnaire, social economic status questionnaire, and 24 h recall for food consumption of lactating women, prepared by the National Institute of Nutrition, Hanoi, Vietnam.

### 2.2. DNA Isolation

DNA was extracted from fecal and breast milk samples thawed at 4 °C. For feces, DNA was isolated following the Repeated Beat-Beating (RBB) procedure that has been included as protocol #5 in the DNA isolation comparison study of Costea et al., 2017 [32]. In short, approximately 0.2 g of feces was used for mechanical and chemical lysis using 750 µL of lysis buffer (500 mM NaCl, 50 mM Tris-HCl (pH 8), 50 mM EDTA, 4% SDS) with 0.5 g added 0.1 mm zirconia beads. Nucleic acids were precipitated by the addition of 200 µL of 10 M ammonium acetate, followed by one volume of isopropanol. Subsequently, DNA pellets were washed with 70% ethanol. Further purification of DNA was performed using the QIAamp DNA Mini Kit (Qiagen, Hilden, Germany). Finally, DNA was dissolved in 200 µL Tris/EDTA buffer.

For DNA isolation from breast milk, 500 µL breast milk was used for mechanical and chemical lysis using 500 µL lysis buffer with 0.5 g added 0.1 mm zirconia beads in combination with a phenol/chloroform extraction. Further purification of DNA was performed using the Awoga Mag Mini DNA isolation kit (LGC genomics, Hoddesdon, UK). Finally, DNA was dissolved in 50 µL elution buffer, and its purity and quantity were checked spectrophotometrically (Qubit 2.0 Fluorometer, Thermo Fisher Scientific, Waltham, MA, USA).

### 2.3. PCR Amplification and 16S rRNA Gene Illumina Sequencing

Barcoded amplicons from the V3-V4 region of 16S rRNA genes were generated using a 2-step PCR (see library PCR below for a description of second PCR step). Universal primers appended with Illumina adaptor sequences were used for the initial amplification of the V3-V4 part of the 16S rRNA gene with the following sequences: ‘5-*TCGTCGGCAGCGTCAGATGTGTATAAGAGACAG*CCTACGGGAGGCAGCAG**’** (broadly conserved bacterial primer 357F underlined); reverse primer, ‘5- *GTCTCGTGGGCTCGGAGATGTGTATAAGAGACAG*TACNVGGGTATCTAAKCC’ (broadly conserved bacterial primer 802R (with adaptations) underlined), appended with Illumina adaptor sequences (in italics). The PCR amplification mixture contained 1 µL fecal sample DNA, 1 µL forward primer (10 µM), 1 µL of reverse primer (10 µM), 14 µL master mix (1 µL KOD Hot Start DNA Polymerase (1 U/µL; Novagen, Madison, WI, USA), 5 µL KOD-buffer (10×), 3 µL MgSO4 (25 mM), 5 µL dNTP mix (2 mM each)), and 33 µL sterile water (total volume 50 µL). The PCR conditions were 95 °C for 2 min, followed by 30 cycles of 95 °C for 20 s, 55 °C for 10 s, and 70 °C for 15 s.

For breast milk, a nested PCR approach was used. The first PCR was performed using the following primers: 338f, 5′-ACTCCTACGGGAGGCAGCAG-3′ and 1061R, 5′-CRRCACGAGCTGACGAC-3′. The PCR amplification mixture contained 2 µL sample DNA, 1 µL forward primer 338f (10 µM), 1 µL reversed primer 1061R (10 µM), 14 µL master mix (1 µL KOD Hot Start DNA Polymerase (1 U/µL; Novagen, Madison, WI, USA), 5 µL KOD buffer (10×), 3 µL MgSO4 (25 mM), 5 µL dNTP mix (2 mM each)), and 32 µL sterile water (total volume 50 µL). The PCR conditions were 95 °C for 2 min, followed by 20 cycles of 95 °C for 20 s, 55 °C for 10 s, and 70 °C for 15 s. The PCR amplicon was subsequently purified using the MSB Spin PCRapace kit (Invitek, STRATEC Molecular GmbH, Berlin, Germany). For the nested PCR, 5 µL of the first PCR product was used. The PCR amplification mixture contained 5 µL sample DNA from first round PCR, 1 µL forward primer 357F (10 µM) (see above), 1 µL reversed primer 802R (10 µM) (see above), 14 µL master mix (1 µL KOD Hot Start DNA Polymerase (1 U/µL; Novagen, Madison, WI, USA), 5 µL KOD-buffer (10×), 3 µL MgSO4 (25 mM), 5 µL dNTP mix (2 mM each)) and 29 µL sterile water (total volume 50 µL). The PCR conditions were 95 °C for 2 min, followed by 25 cycles of 95 °C for 20 s, 55 °C for 10 s, and 70 °C for 15 s.

For both sample types, the final PCR amplicons were subsequently purified using the MSB Spin PCRapace kit (Invitek, STRATEC Molecular GmbH, Berlin, Germany). The concentration and quality were subsequently checked with a Qubit fluorometer. The purified PCR products were shipped to BaseClear BV (Leiden, The Netherlands) and used for the second (library) PCR in combination with sample-specific barcoded primers. The PCR products were checked and quantified on a Bioanalyzer (Agilent, Santa Clara, CA, USA), followed by multiplexing, clustering, and sequencing on an Illumina MiSeq (San Diego, CA, USA) with the paired-end (2×) 300 bp protocol and indexing. The sequencing run was analyzed with the Illumina CASAVA pipeline (v1.8.3), with demultiplexing based on sample-specific barcodes. The raw sequencing data produced was processed, removing the sequence reads of too low quality (only “passing filter” reads were selected) and discarding reads containing adaptor sequences or PhiX control. A quality assessment of the remaining reads was performed using the FASTQC quality control tool version 0.10.0. (http://www.bioinformatics.babraham.ac.uk/projects/fastqc/).

### 2.4. 16S rRNA Gene Sequence Analysis and Statistics

16S sequences were analyzed using a workflow based on QIIME 1.8 [33]. OTU clustering, taxonomic assignment, and reference alignment were performed with the pick_open_reference_otus.py workflow script of QIIME, using uclust as the clustering method (97% identity) and GreenGenes v13.8 as the reference database for taxonomic assignment. Reference-based chimera removal was performed with Uchime [34]. The RDP classifier version 2.2 was performed for taxonomic classification [35]. Samples with fewer than 10,000 reads after the removal of chimeras and singletons (reads in OTUs with only a single sequence) were removed. The microbiota transfer from mother to infant was studied using OTUs picked at 99% identity, using “fraction of shared OTUs” as the distance measure, after down-sampling to the lowest common number of reads using the rarefaction workflow implemented in QIIME. For the analysis of specifically *Enterobacteriaceae* sequences, all reads assigned to an OTU classified as “*Enterobacteriaceae*” were taken, and used as input for a new QIIME analysis, with OTUs picked at 99% identity.

Statistical tests were performed as implemented in SciPy (https://www.scipy.org/), downstream of the QIIME-based workflow. We tested for between-group differences in alpha diversity (PD_whole tree) and beta diversity (weighted UniFrac; for each subject in a group, the average distance to all subjects in another group was calculated) with the non-parametric Kruskal–Wallis test with Dunn’s post hoc test, as implemented in Graphpad Prism 5.01 (San Diego, CA, USA). Between group-differences of single taxa were assessed using non-parametric tests. For comparisons of more than two groups, the non-parametric Kruskal–Wallis test with FDR correction was applied, unless otherwise stated. The comparisons of targets of our primary interest (*Bifidobacteriaceae* and *Enterobacteriaceae*) were not corrected for multiple testing.

To compare the global difference in microbiota compositions between groups and to assess associations with breast milk nutrients, we performed principal component analysis (PCA) and multivariate redundancy analyses (RDAs) in Canoco version 5.12, using the default settings of the analysis type “Unconstrained” and “Constrained”, respectively [36]. The relative abundance values of OTUs or genera were used as the response data, and the metadata as the explanatory variable. For visualization purposes, families or genera, rather than OTUs, were plotted as supplementary variables. Variation explained by the explanatory variables corresponds to the classical coefficient of determination (R2) and was adjusted for degrees of freedom (for explanatory variables) and the number of cases. Canoco determines RDA significance by permutating (Monte Carlo) the sample status. Per sample set, confounding factors were first identified by RDA. Statistically significant confounders were included as covariates in subsequent analyses. Hence, partial RDA was employed to correct for covariance where relevant, and covariates were first fitted by regression and then removed from the ordination.

### 2.5. Dietary Intake Analysis

Dietary information was collected via a 2 × 24 h recall for food consumption of lactating women, provided by the National Institute of Nutrition, Ha Noi, Vietnam. The nutrient composition of the diet was estimated using the Vietnamese food composition table [37,38].

### 2.6. Blood Parameters

For vitamin B12, non-fasting venous heparin-anticoagulated blood samples were taken from the mothers on the day of breast milk collection. Plasma was isolated by centrifugation at 2500× *g* for 10 min, and frozen and stored at −80 °C. The transport to the University Medical Center Groningen (Groningen, The Netherlands) took place on dry ice, where samples were stored at −20 °C until analysis. Plasma vitamin B12 was measured with an Electro-Chemiluminescence immunoassay (Roche Diagnostics, Basel, Switzerland) on various days. The inter-assay CVs were 4.7% and 3.3% at plasma vitamin B12 concentrations of 295 and 564 pmol/L, respectively. Hemoglobin was measured in whole blood with HemoCue Hb 201 (HemoCue AB, Ängelholm, Sweden). Serum ferritin was measured in heparin plasma using an immunoturbidimetric assay (Roche, Cobas c 601, ELISA. Ramco Laboratories, Inc., Houston, TX, USA).

### 2.7. Breast Milk Component Analysis

Breast milk Ca, I, K, Mg, Na, Se, Zn, Cu, and Fe were analyzed by ICP-MS (European Laboratory of Nutrients, Bunnik, The Netherlands). The breast milk samples were thawed at room temperature. Subsequently, they were diluted 1:20 with an alkaline solution of 0.1% ammonia, 0.03% Triton-X-100, and 0.03% EDTA. After dilution, the samples were analyzed by ICP-MS. Quality was ensured by the use of control cards by the NEN norm 6603 internally. By use of (pooled) breast milk samples, intra-assay CVs at low and high levels of <12% for potassium, calcium, sodium, magnesium, zinc, iron, copper, iodine, and selenium were found. Their inter-assay CVs were <15%.

Vitamin B12 was analyzed using Immulite 1000 vitamin B12 assay (Siemens Healthcare, Munich, Germany), as previously described [39]. Thawed breast milk samples were centrifuged at 3300× *g* and 4 °C. A 200 µL aliquot of the whey fraction was used for the analysis. Inter-assay and intra-assay CVs were <15% according to the manufacturer. All samples were analyzed on a single day.

Vitamin D (ARA; antirachitic activity) was determined by liquid-chromatography-MS/MS, as previously described [40].

Fatty acids were determined by capillary gas chromatography, as previously described, and values were expressed as gram percent of total fatty acids [41,42].

Mother secretor status and breast milk 2′FL concentration were determined using 1D ^1^H NMR spectroscopy, as previously described [31].

For the multivariate analysis on breast milk components, the data of each component were first scaled by dividing each value by the average of that component.

## 3. Results

### 3.1. Demographics of the Cohort

Questionnaires (dietary, health, and social economic status) and fecal and breast milk samples were taken from subjects from five different regions in Vietnam (Appendix A). Of all participants, 69.3% provided exclusive breastfeeding during the sampling period. Some participants did not fill out the food questionnaire (7.9%). The remaining 22.8% provided mixed feeding of which breastmilk was a significant part of the diet. In most of these cases (17.8%), the feeding was mixed with formula. In 2.0% of the cases, other liquids, e.g., water or juice, were added to the diet; in 2.0%, mashed solids were already added to the diet; and in 1.0%, formula as well as other liquids were added [31].

### 3.2. Quality Control and Overall Microbiota Composition

16S rRNA gene Illumina sequencing of the V3-V4 region was applied to determine the bacterial composition of the milk and fecal samples. A total of 251 samples (78 infant feces, 82 mother feces, and 91 breast milk) passed all quality control checks, with an average number of reads per sample of 36,334, with a standard deviation of 15,352. The principal component analysis (PCA) showed a clear sample clustering of the age groups (infants, mothers) and sample niches (feces, breast milk), as expected (Appendix A). The infant fecal samples were typically characterized by *Bifidobacterium*, while the mother fecal samples were characterized by adult-like taxa such as *Ruminococcus* and *Blautia*. The breast milk samples were associated with typical skin-associated bacteria such as *Corynebacterium* and *Propionibacterium*. In line with the expectations, within-sample (α-)diversity was different between the three sample types, i.e., low in infant feces and breastmilk, but high in maternal feces (Appendix A). An overview of the overall microbiota composition for all sample types is included in Appendix A.

### 3.3. Associations between Distinct Geographical Areas and Infant Fecal Microbiota Composition

The redundancy analysis (RDA) revealed a significant association between geography and relative abundance at the infant fecal microbiota genus level (variation explained 5.9%, *p* = 0.002) (Figure 1). Ha Noi infants showed a distinct gut microbiota composition and were characterized by higher relative abundances of, e.g., *Klebsiella*, *Staphylococcus*, and *Citrobacter*. The separation of Ha Noi infants from the other infants was mainly on the horizontal RDA axis (the most important axis). On the vertical axis, both cities were separated from the rural regions, although Ho Chi Minh was more similar to the rural regions than to the city of Ha Noi. The rural regions were characterized by higher relative abundances of *Bifidobacterium*. Among the rural regions only, there was no statistically significant association between the infant fecal microbiota composition and region as assessed by RDA. Mixed feeding with formula versus exclusive breastfeeding did not significantly impact the infant fecal microbiota composition in this dataset.

Within-sample diversity (Faith’s PD_whole_tree metric) was significantly different between regions (*p* = 0.048, Kruskal–Wallis test) (Appendix A). Within-sample diversity was lowest in Ha Noi and highest in Tien Giang, with diversity in Ha Long Bay being relatively low compared to the other rural regions, although the individual pairwise comparisons were not significant (*p* > 0.05, Dunn’s multiple comparison test). The characteristic genera and within-sample diversity for the individual regions are listed in Appendix A.

### 3.4. Phylogenetic Analyses of the Enterobacteriaceae as Present in the Infant Microbiota and Maternal Breast Milk Samples

As the Ha Noi infant fecal microbiota composition was characterized by potentially pathogenic *Enterobacteriaceae* such as *Klebsiella* and *Citrobacter*, in-depth phylogenetic analyses (OTU clustering at 99%) and a subsequent multivariate analysis was performed on all infant and breast milk samples. The association between *Enterobacteriaceae* OTU relative abundance and region was significant (Figure 2A, *p* = 0.012, explained variation 5.7%). The number of different *Enterobacteriaceae* OTUs (indicating diversity/richness) was strikingly higher in most infant fecal samples from urban Ha Noi (Figure 2B and Figure 3). Similarly, *Enterobacteriaceae* were also associated most strongly with breast milk samples from Ha Noi (Figure 2C, explained variation 8.6%, *p* = 0.002), and the diversity (number of different OTUs) of *Enterobacteriaceae* was much higher in most Ha Noi breast milk samples as compared to the other regions (Figure 2D and Figure 3).

The bivariate analysis on *Enterobacteriaceae* diversity/relative abundance and region confirmed that both abundance and diversity were relatively high in infants in (urban) Ha Noi. This was also the case for maternal fecal and breast milk samples. In breast milk samples, *Enterobacteriaceae* relative abundance and diversity was also relatively high in rural Phu Tho (Figure 3), but it was associated less with *Klebsiella* and *Citrobacter* compared to Ha Noi (as indicated by RDA; Appendix A). *Enterobacteriaceae* OTU dissimilarity (β-diversity/distance between samples, as a measure of whether samples within a region contain more similar or dissimilar *Enterobacteriaceae* spp.) was highest between infants within Phu Tho, followed by Ha Noi, while in breast milk, it was highest in Ho Chi Minh, followed by Ha Noi (Figure 3). In summary, the high *Enterobacteriaceae* relative abundance (including pathobionts *Klebsiella* and *Citrobacter*) and α-diversity were most striking in both infant fecal samples and breast milk samples from (urban) Ha Noi.

### 3.5. Associations between Geography and Breast Milk Nutrient Composition

RDA was used to reveal associations between geographic area and breast milk composition. The association between region and the normalized concentration of the components was significant, and the region explained 16.7% of the variation (*p* = 0.002) (Figure 4). Mothers in Ha Long Bay showed the most distinct breast milk profile as compared to the other regions. The most important drivers for this separation were eicosapentaenoic acid (EPA), selenium, vitamin B12, and docosahexaenoic acid (DHA), which appeared in higher concentrations in the breast milk of mothers living in Ha Long Bay. As the dietary intake of the lactating mothers was determined (Table 1), we conclude that these nutrients could be associated with the consumption of saltwater fish that is characteristic for the Ha Long Bay diet. Interestingly, mothers in Ha Noi had relatively high concentrations of α-linoleic acid, linoleic acid, iron, and zinc in their breast milk. From the food questionnaires, it could not exactly be discerned from which dietary sources these nutrients could originate. Most obvious was the relatively low intake of plant-based fat in Ha Noi, in combination with a relatively high intake of animal protein. There was no correlation between iron concentration in the breast milk and mother plasma ferritin concentration or hemoglobin concentration.

### 3.6. Assessment of Maternal Transmission of Fecal and/or Breast Milk Microbiota to the Infant Fecal Microbiota

To assess maternal transmission of fecal and/or breast milk microbiota (seeding) to the infant fecal microbiota, OTUs were clustered at the 99% identity level. The percentage of shared OTUs between mother samples and infant samples was calculated, and this was compared between mother–infant pairs (within families) and mother–random-infant pairs (between families) for each region. Both for mother fecal samples and breast milk samples, the number of shared OTUs was low and was not different for mother–infant pairs compared to mother–random-infant pairs, in any of the regions. Hence, these analyses did not support mother-specific microbiota seeding signatures in these infants of approximately 40 days old. Nevertheless, in the *Enterobacteriaceae*-specific analysis, it was striking that high relative abundances and diversity of *Enterobacteriaceae* were found in both breast milk and infant fecal samples from Ha Noi (Figure 2). Remarkably, as shown in Figure 3, the relatively high abundance and diversity of *Enterobacteriaceae* in Phu Tho breast milk was not reflected in the microbiota composition of infant fecal samples in Phu Tho, but the dissimilarity in *Enterobacteriaceae* OTUs was relatively high between those samples, which indicates that infants within Phu Tho carry relatively different *Enterobacteriaceae* communities compared to each other. Of note, both *Enterobacteriaceae* relative abundance and diversity followed the same region trend for infant and mother fecal microbiota, while this was less apparent for the breast milk samples (Figure 3).

### 3.7. Associations between Breast Milk Nutrients and Infant Microbiota Composition

As the breast milk nutrient composition was significantly different between the regions, the potential association between the nutrients and infant fecal microbiota composition was assessed. Figure 5 shows a PCA of the infant fecal microbiota composition, with the breast milk nutrients plotted as supplementary variables. Interestingly, iron was positively associated with *Klebsiella* and negatively associated with *Bifidobacterium*, while EPA and DHA were positively associated with *Bifidobacterium*. These associations were indicative only, as the single nutrient concentrations did not significantly associate with infant fecal microbiota composition after correction for multiple testing. The variation explained by each breast milk nutrient variable and corresponding *p*-values (unadjusted and FDR adjusted) are summarized in Table 2. The breast milk components concentrations are included in Appendix A.

### 3.8. Associations between Mother’s Secretor Status and Infant Fecal Microbiota Composition

Mother secretor status had a significant effect on infant fecal microbiota composition (variation explained 1.0%, *p* = 0.046) (Figure 6). Secretor status was associated with higher relative abundances of, e.g., *Streptococcus* and *Enterobacter*, while infants of non-secretors were associated with higher relative abundances of, e.g., *Collinsella* and *Lactobacillus* (lactobacilli “pre-2020 nomenclature”); *Bifidobacterium* was not associated with either group. Of note, the relative abundance of *Enterobacteriaceae* was not significantly different between infants of secretors/non-secretors in the bivariate analysis (*p* = 0.72). Secretor status did not have an association with breast milk microbiota composition in this dataset.

## 4. Discussion

Infant gut microbiota colonization and development is influenced by many extrinsic factors such as delivery and feeding mode, maternal diet and lifestyle, and other environmental factors [4,5,6,7]. In this study we revealed the associations between various geographic regions in Vietnam (urban/rural areas, with different socioeconomic status) and maternal-specific differences (diet and breast milk composition) with vaginally born infant fecal microbiota.

A strong effect of geography on infant gut microbiota composition was found in the current study. The association was mostly driven by samples from Ha Noi, in which the relative abundance of potentially detrimental pathobionts, such as *Klebsiella* and *Citrobacter*, was strikingly high as compared to other regions in this study. Despite the different characteristics between the rural areas (Table 1, Appendix A), no overall statistically significant difference in infant gut microbiota composition was found between these regions. On the basis of the available study data, it is difficult to explain the relatively high abundance and diversity of *Enterobacteriaceae* in Ha Noi. Although OTU seeding between mother and infant pairs (within regions) was not supported by our analysis, *Enterobacteriaceae* relative abundance and diversity in breast milk were also found to be relatively high in Ha Noi. In addition, breast milk iron content was relatively high in Ha Noi, which may have promoted the growth of potentially pathogenic *Enterobacteriaceae*, as it is well known that enteropathogens can be stimulated by iron [43]. Although maternal diet is an important determinant of breast milk composition, both for nutrients and microbiota, it is difficult to elucidate where the iron originates from [12,14,23]. We did not find a correlation between iron concentration in the breast milk and mother plasma ferritin concentration or hemoglobin concentration. The concentration of zinc was also relatively high in breast milk samples in Ha Noi. The breast milk concentration of both minerals is thought not to be affected much by maternal diet mineral content, as it concerns well-regulated homeostatic processes [14]. Nevertheless, as associations of breast milk iron and zinc with iron supplementation and meat intake have been reported [44], our observations suggest that mothers in Ha Noi ingested higher levels of iron and zinc as compared to the other regions. The origin of this additional iron may also be polluted ground water around Ha Noi [45], or a higher meat intake (westernized diet). Animal protein intake in Ha Noi was relatively high, in combination with a relatively low plant-based fat consumption. The association with *Enterobacteriaceae* was not found in Ho Chi Minh City, where animal protein intake was also high, but perhaps the diet in Ho Chi Minh City was generally more balanced, as reflected by a much higher intake of plant-based fat. Hygiene practices in meat processing and cooking could also play a role in enteropathogen transmission, but potential differences in these practices between the study populations in the two cities were not investigated. Another potential factor in the *Enterobacteriaceae* abundance is the widespread use of over the counter antibiotics, or inappropriate prescription, in Vietnam, which can select for resistant intestinal *Enterobacteriaceae* [46,47]. Maternal medication use during the 4 weeks before sampling was an exclusion criterion. However, information on historic use of antibiotics was not retrieved for this study population. Therefore, it is unclear whether previous maternal antibiotic use in Ha Noi may have played a role. Given the low intake of plant-based fat in Ha Noi, the fat composition of the maternal diet may have contributed to the differences observed in infant fecal microbiota composition. A recent study indicated that dietary fat, specifically fish-derived fat, is associated with gut microbiota composition, in particular with generally considered beneficial typical butyrate producers [48]. In addition, the breast milk omega-3 fatty acid concentration was relatively low in Ha Noi compared to Ho Chi Minh City, while supplementation of these fatty acids has been reported to have beneficial (e.g., bifidogenic) effects [49,50].

Breast milk nutrient composition is an important factor in shaping the infant gut microbiota. In addition to iron and zinc, as mentioned above, we also measured the level of a number of additional breast milk minerals, as well as fatty acids, vitamins, and electrolytes. In addition, the mother secretor status was determined based on HMO structure analysis in the breast milk samples [31]. The RDA showed a clear association of breast milk components with region. Most striking was the breast milk nutrient composition in Ha Long Bay, which was associated with the fatty acids EPA and DHA, and the mineral selenium. Ha Long Bay is the only region in our dataset where salt water fish consumption is high (coastal region), and the outcomes are in line with recent studies that showed higher levels of DHA in breast milk from coastal regions, while arachidonic acid levels were less variable [19,51]. In general, DHA and EPA were not among the nutrients that were associated with infant gut microbiota composition the strongest, but breast milk samples with relatively high DHA and EPA concentrations tended to be associated with, e.g., *Bifidobacterium* relative abundance in the infant microbiota samples, in line with the reported prebiotic effect of omega-3 fatty acids in adults [30,31]. The RDA on breast milk nutrients showed that the Ha Long Bay samples were most distinct, but to what extent this shapes the infant fecal microbiota is difficult to discern, as the fecal microbiota composition of Ha Long Bay infants was not much different from the other rural provinces.

The breast milk HMO composition as well as the secretor status and Lewis blood group of the mothers was determined based on an HMO structure analysis in the (corresponding four groups’) breast milk samples [31]. We showed that secretor status was not significantly associated with either breast milk microbiota or infant fecal microbiota composition. However, when region was included as a covariate, a borderline statistically significant association was found with secretor status and infant fecal microbiota composition. This suggests that secretor status impacted infant gut microbiota composition, but that other regional factors were more important, such as the iron concentration in breast milk. When region was included as a covariate (ruling out the effect of region), the breast milk iron concentration was not significantly associated with infant gut microbiota composition, indicating a strong link of breast milk iron with Ha Noi in particular. Although the average iron level in Ha Noi breast milk was relatively high, it should be noted that it was still within the normal range of 0.2–0.8 mg/L in all regions [52]. On the basis of 14 reference HMO structures, the levels of fucosylated HMOs have been previously estimated [31]. Interestingly, the lowest average fucose-containing HMO levels were observed in Ha Noi, and the highest in Ha Long Bay and Ho Chi Minh. Whether the low level of fucosylated HMOs in combination with relatively high iron levels in the breast milk of Ha Noi mothers gave rise to a higher *Enterobacteriaceae* load in both breast milk and the infant fecal microbiota remains to be elucidated. Previous studies have reported inconsistent findings regarding solely mother secretor status and association with the gut microbiota composition of their infants. Some studies observed a significant link [53,54,55], e.g., a positive association with *Bifidobacterium*, but others did not [56,57]. Notably, our findings also support the idea that maternal secretor status is just one of the many factors that modulate infant gut microbiota composition, and that other population-specific factors also play a role. In the current study population, the ratio between secretor mothers and non-secretor mothers was different compared to observations in Western studies. The number of non-secretor individuals was higher in Vietnam (39.6%) as compared to Western societies (21%), with regional differences [31]. A recent study has shown that maternal diet intervention can alter breast milk HMO and microbiota composition [58], highlighting the potential of modulating the maternal diet in order to target the infant microbiota.

## 5. Conclusions

In summary, geography and (corresponding) diet and lifestyle clearly impact infant fecal microbiota composition in Vietnam. A consistent distinction between the rural and urban regions was not observed, but the urban region of Ha Noi was clearly distinct from the other regions, including urban Ho Chi Minh. We cannot be conclusive about the origin of this distinction, but our analyses point in the direction of diet. Correlation analyses indicated that specific breast milk components could be associated with both beneficial and potentially harmful infant microbiota members, which could give guidance to beneficially steer infant microbiota composition and development through maternal (dietary and lifestyle) interventions.

## Figures and Tables

**Figure 1 nutrients-15-04802-f001:**
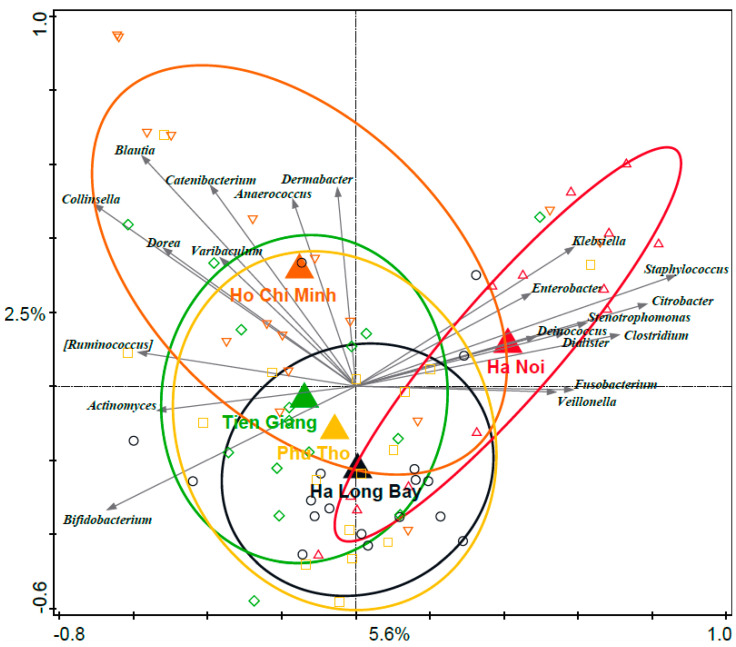
RDA of infant fecal microbiota relative abundance on the genus level and geography. Large triangles are centroids of the sample groups (regions), while the other symbols indicate individual samples. Ellipses are the 66% quantile of the approximated 2D-normal density distribution function for each region. Grey arrows are the 20 best-fitting genera. Variation explained by region was 5.9%, *p* = 0.002.

**Figure 2 nutrients-15-04802-f002:**
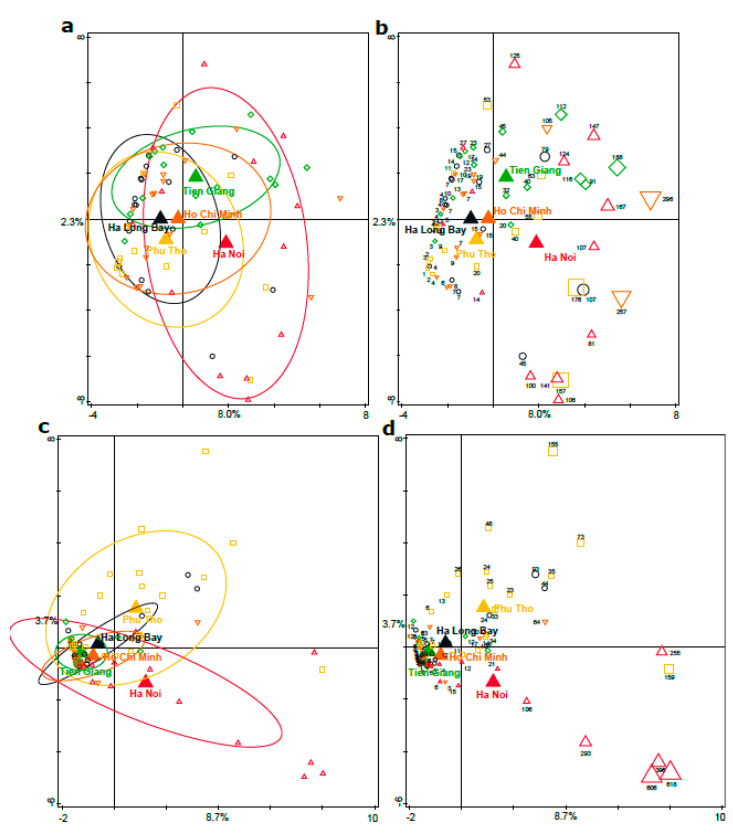
RDA of *Enterobacteriaceae* relative abundance on the OTU level and geography. (**a**) Infant feces biplot. (**b**) Infant feces attribute plot; variation explained by region was 5.7%, *p* = 0.012. (**c**) Milk biplot. (**d**) Milk attribute plot; variation explained by region was 8.6%, *p* = 0.002. Filled triangles are centroids of the sample groups (regions), while the other symbols indicate individual samples. Ellipses are the 66% quantile of the approximated 2D-normal density distribution function for each region. The number with each sample indicates the number of different *Enterobacteriaceae* OTUs for that sample, also reflected by the symbol size.

**Figure 3 nutrients-15-04802-f003:**
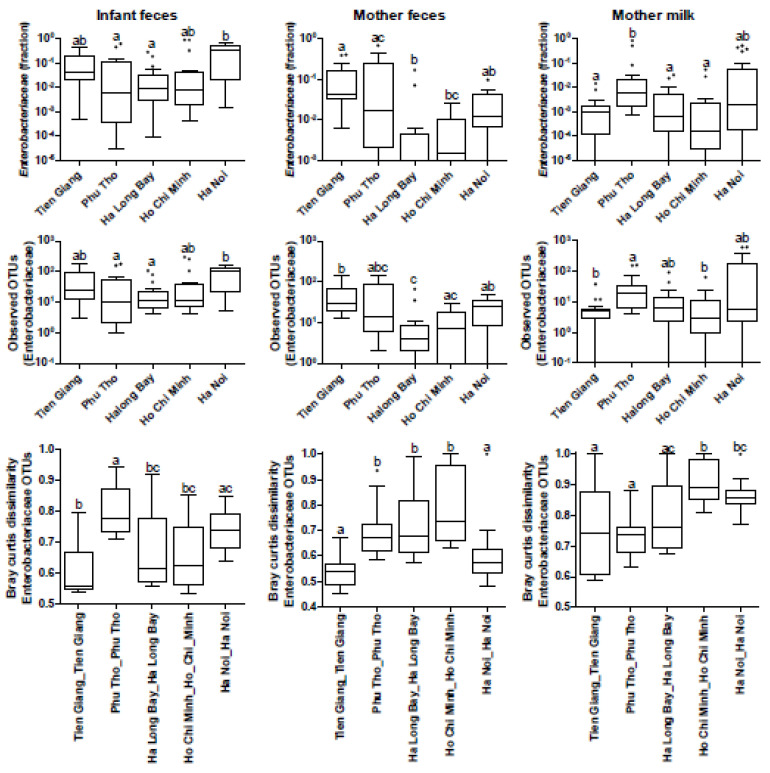
*Enterobacteriaceae*-specific analysis per sample type and geography. (**Top panels**) present *Enterobacteriaceae* relative abundance: infant; *p* = 0.0046, mother; *p* < 0.0001, and milk; *p* = 0.0005. (**Middle panels**) present the number of observed *Enterobacteriaceae* OTUs (diversity): infant; *p* = 0.0112, mother; *p* < 0.0001, and milk; *p* = 0.0011. (**Bottom panels**) present the phylogenetic distance (Bray–Curtis dissimilarity based on *Enterobacteriaceae* OTUs only; β-diversity) between samples within each region: infant; *p* < 0.0001, mother; *p* < 0.0001, and milk; *p* < 0.0001. Boxplots are displayed as Tukey boxplots. The Kruskal–Wallis test with Dunn’s post hoc test was used to assess differences between regions. Whiskers that do not have a letter in common are significantly different (*p* < 0.05).

**Figure 4 nutrients-15-04802-f004:**
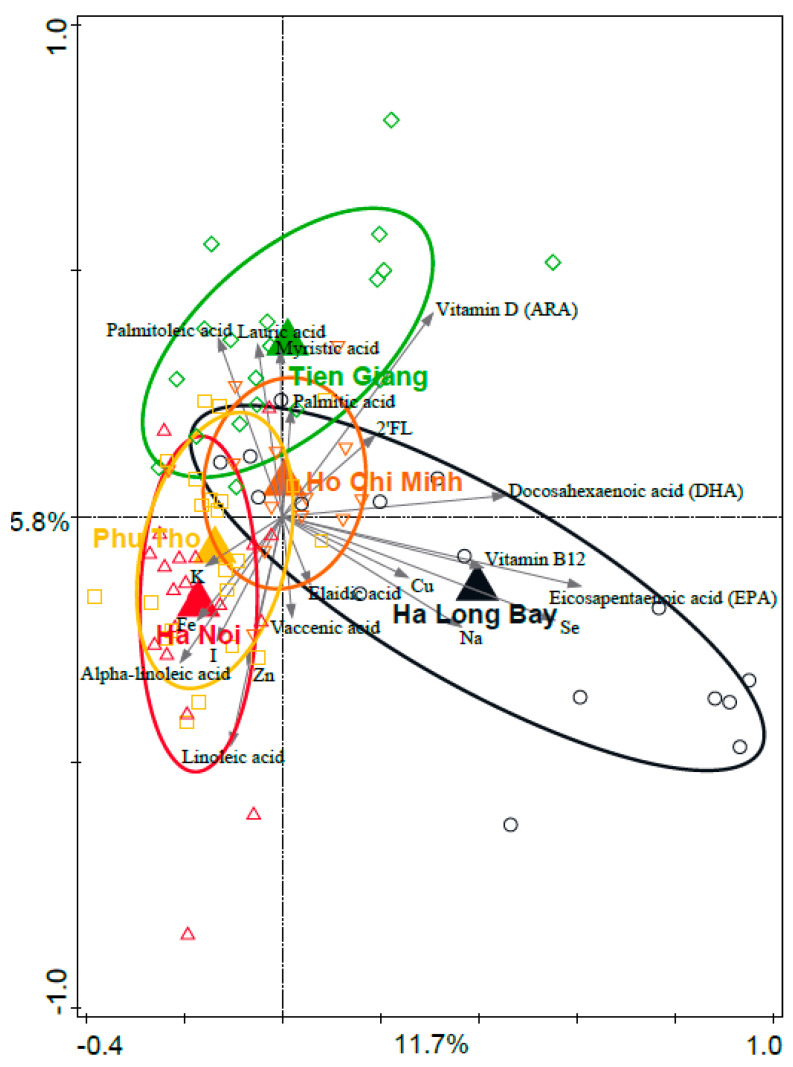
RDA of breast milk nutrients (normalized) and geography. Large triangles are centroids of the sample groups (regions), while the other symbols indicate individual samples. Ellipses are the 66% quantile of the approximated 2D-normal density distribution function for each region. Grey arrows are the 20 best-fitting nutrients. Variation explained by region was 16.7%, *p* = 0.002.

**Figure 5 nutrients-15-04802-f005:**
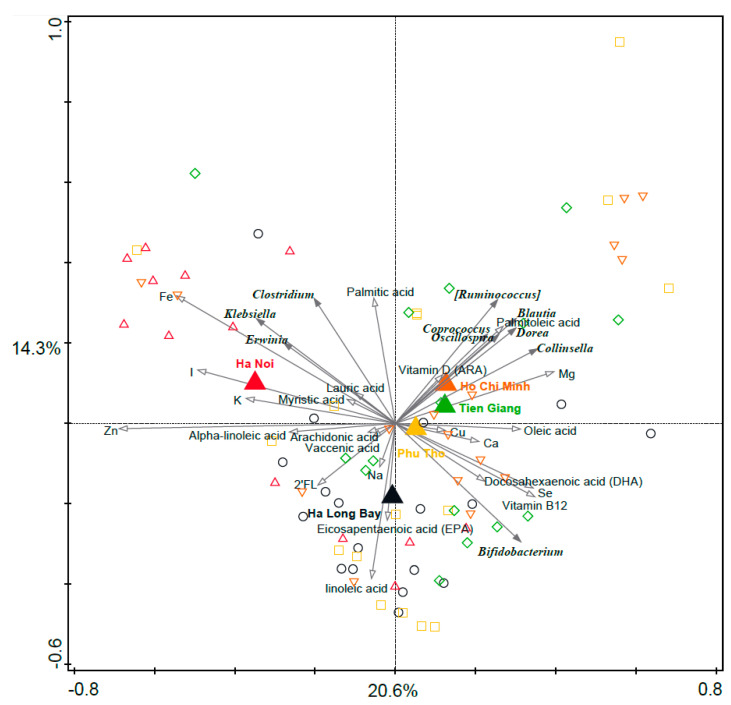
PCA of infant fecal microbiota relative abundance on the genus level, with breast milk nutrients plotted as supplementary variables. Genera were used as response data, and breast milk nutrients (normalized) and regions were plotted as supplementary variables. Breast milk nutrients explain 4.8%, and together with region the supplementary variables explain 9.1% of the variation. Large triangles indicate region centroids and are plotted as supplementary information. Filled arrows show the 10 best fitting genera. Open arrows show the 23 compounds that were determined in the breast milk samples.

**Figure 6 nutrients-15-04802-f006:**
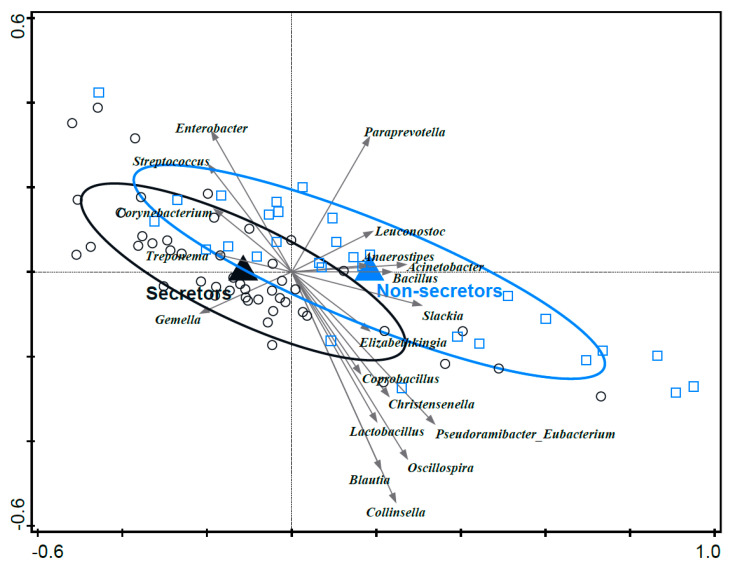
RDA of infant fecal microbiota relative abundance on the genus level and mother secretor status. Genera were used as the response data, and secretor status was used as the explanatory data. Covariance attributable to region was first fitted by regression and then partialled out (removed) from the ordination. Ellipses are the 66% quantile of the approximated 2D-normal density distribution function for each region. Grey arrows are the 20 best-fitting genera. Variation explained by secretor status was 1.0%, *p* = 0.046.

**Table 1 nutrients-15-04802-t001:** Dietary characteristics of the regions, specified for total energy intake, protein intake, and lipid intake.

Region	Typical Diet	Energy (kcal)	Protein (g)	Animal Protein (g)	Fat (g)	Plant-Based Fat (g)
Tien Giang	Freshwater fish	2022 ^a^	87 ^b^	52 ^b^	52 ^b^	10 ^ab^
Phu Tho	Rice	1677 ^c^	65 ^a^	27 ^a^	36 ^a^	12 ^a^
Ha Long Bay	Saltwater fish	1711 ^ac^	72 ^ab^	32 ^a^	36 ^a^	12 ^a^
Ha Noi	Westernized	2053 ^b^	93 ^b^	60 ^b^	50 ^b^	4 ^b^
Ho Chi Minh City	Westernized	2214 ^b^	99 ^b^	59 ^b^	74 ^c^	23 ^ab^

Within columns, values that do not have a letter in common are significantly different (*p* < 0.05; Kruskal–Wallis test with Dunn’s post hoc test). The kcal intake is significantly lower in Ha Long Bay compared to Ha Noi and Ho Chi Minh, and lower in Phu Tho compared to Ha Noi, Ho Chi Minh, and Tien Giang. The protein intake is significantly lower in Phu Tho compared to Ha Noi, Ho Chi Minh, and Tien Giang. The animal protein intake is significantly lower in Phu Tho compared to Ha Noi, Ho Chi Minh, and Tien Giang, and lower in Ha Long Bay compared to Ha Noi, Ho Chi Minh, and Tien Giang. The fat intake is significantly lower in Ha Long Bay compared to Ho Chi Minh, and lower in Phu Tho compared to Ha Noi, Ho Chi Minh, and Tien Giang. The plant-based fat intake is significantly lower in Ha Noi compared to Phu Tho and Ha Long Bay.

**Table 2 nutrients-15-04802-t002:** Summary of variation explained by breast milk nutrients on infant gut microbiota composition. The RDA that summarizes the effect of explanatory variables (breast milk nutrients that were measured in this study) was performed on the genus level of infant gut microbiota composition. The nutrients are ordered based on the percentage of variation explained in the infant microbiota composition. Region was not included as a covariate in this analysis.

Breast Milk Nutrient	Explains (%)	*p*-Value	*p*-Value (FDR)
Zinc	3.3	0.012	0.161
Iron	2.7	0.014	0.161
Iodine	2.2	0.068	0.355
Potassium	2.2	0.072	0.355
Magnesium	2.0	0.108	0.355
Vitamin B12	1.9	0.100	0.355
Selenium	1.9	0.104	0.355
Palmitoleic acid	1.7	0.204	0.587
Copper	1.6	0.242	0.618
Docosahexaenoic acid (DHA)	1.5	0.294	0.619
Oleic acid	1.4	0.296	0.619
Linoleic acid	1.3	0.396	0.715
2’FL	1.3	0.404	0.715
Vaccenic acid	1.2	0.484	0.795
Vitamin D (ARA)	1.1	0.528	0.810
Alpha-linoleic acid	1.0	0.696	0.852
Lauric acid	1.0	0.752	0.852
Palmitic acid	0.9	0.722	0.852
Calcium	0.9	0.770	0.852
Sodium	0.9	0.764	0.852
Eicosapentaenoic acid (EPA)	0.9	0.778	0.852
Myristic acid	0.9	0.820	0.857
Arachidonic acid	0.6	0.978	0.978

## Data Availability

Appendix A accompanies this paper and is available online. The raw sequencing data generated and analyzed during the current study will be made available in an online public repository after manuscript acceptance.

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
