# Peer review of "Mothers’ Breast Milk Composition and Their Respective Infant’s Gut Microbiota Differ between Five Distinct Rural and Urban Regions in Vietnam"

_nutrients, 2023, doi:10.3390/nu15224802_

Round 1

Reviewer 1 Report

Comments and Suggestions for Authors

The subject of the research of the submitted manuscript Impact of geography and mum on the infant’s gut microbiota composition in Vietnam, focuses on the analysis of an infant's intestinal microbiota with the aim of discovering how dependent it is on mother milk microbiome composition in the context of different gastronomic habits, including cohorts from 5 regions in Vietnam. The analysis of intestinal microbiota is among the world´s most attractive scientific topics.

My comments are as follows:

I disagree with the title. Rephrase it to better reflect the study.

Can the authors justify or assume a reason for the higher prevalence of the specific compounds in the breast milk of mothers from the 2 regions listed in the Abstract?

Not geography, but specific exogen factors are the determinants that influence these correlations. please specify

These are no appropriate keywords geography; urban; rural; Vietnam-these do not reflect the target search content- REVISE

Line 30-delete currently, too much wording: human gut and systemic health and wellbeing

Lines 47-49 reformulate the sentence to better convey the main information to the reader

Line 50 here and elsewhere, authors use some unusual words/terms not common among the terminology of this field:  tuned; before in text: foundations, intimate

Please specify the number of breast milk samples for analysis i.e the number of breast milk-infant feces pairs

-incorrect referencing, check the journal propositions: van Leeuwen, S. S. et al. Regional variations in human milk oligosaccharides in Vietnam suggest FucTx activity besides FucT2 and FucT3.

Line 99-100: state the approval date and permission of the ethics document's number

Line 178 16. S rRNA correct to 16S rRNA

Line 179 correct to QIIME

Line 277-278 Corynebacterium and Propionibacterium are typical representatives of the human skin microbiome. You should argue better that their high relative abundances among breast milk bacterial populations are not due to the contamination by skin species from the mother`s hands or breasts.

Line 285 check the error

Figure 1. Interesting, but difficult to read.

I would love to see a graphical showing for example average abundance of OTUs at the class level.

Figure 3. Same impression as for Fig.1. hard to read…can you better mark the different regions at x axes

When commenting on a diversity specify whether is it alpha or beta

Line 339-351 improve; what is the significance of stating the specific region? The results should be commented on by keeping in mind that are being considered from the global scientific population will read the paper. If the aim is to focus on the differences in microbiota composition that for example arise from the hypothesis that the mothers coming from urban regions probably have a different microbiome composition than those coming from rural regions the results should be presented in this context ...

Results section—all subtitles are written as conclusions. This must be modified

Line Table 1 –not regions but typical food of the region

Author Response

We would like to thank the reviewer for the evaluation of the manuscript and the very valuable points raised.

Please find below our sincere efforts to address all. The adjusted version of the manuscript has been uploaded.

  • The title has been rephrased to better reflect the study. Please note: adjusted title not adjusted in the “Citation” in the left margin of the front page.
  • Most striking was the breast milk nutrient composition in Ha Long Bay, which was associated with higher concentrations of the fatty acids EPA and DHA, and the mineral selenium. Ha Long Bay is the only region in our data set where saltwater fish consumption is high (coastal region) and the outcomes are in line with recent studies that showed higher levels of DHA in breast milk from coastal regions. Our observations suggest that mothers in Ha Noi ingested higher levels of iron and zinc as compared to the other regions. The origin of this additional iron and zinc may be associated with meat consumption (westernized diet) as reported before as animal protein intake in Ha Noi was relatively high, in combination with a relatively low plant-based fat consumption. Polluted ground water around Ha Noi could also have impacted the iron and zinc levels. The reasoning is included in the Discussion section.
  • Dietary habits/gastronomic habits are indeed important exogen factors reflecting differences between geographic areas. The title and phrasing in the Abstract have been adjusted to address this fair point.
  • The key words have been adjusted to better reflect the aim, focus and outcomes of the study.
  • Line 30: “currently” has been deleted, sentence has been shortened.
  • Lines 47-49: The sentence has been reformulated to better convey the main information to the reader.
  • Line 50, elsewhere: The words/terms that are not common in this research field have been adjusted (e.g. “tuned” line 50, “foundations” line 41, “intimate” line 32).
  • The number of breast milk samples and breast milk-infant feces pairs have been added to the Material & Methods section, lines 108-111.
  • The references have been adjusted in line with the authors guidelines (ACS-style).
  • Line 99-100: The approval date and permission of the ethics document, 18 July 2013, has been added.
  • Line 178: corrected.
  • Line 179: corrected.
  • Lines 277-278: Although in the breast sampling protocol the medical doctors were instructed to first swab the skin before milk collection, “contamination” with skin bacteria cannot be excluded. However, Corynebacterium and Propionibacterium are commonly isolated from breast milk, e.g. as described by Gonzalez et al 2021, Front. Microbiol.
  • Line 285: The ERROR (due to an incorrect reference to a Figure) has been resolved (the right Figure number has been included), also at other positions in the manuscript.
  • Figure 1 has been adjusted to improve the readability. The new version has been uploaded.
  • Figure 3 has been adjusted to improve the readability. The new version has been uploaded. An additional (supplementary) Figure (Supplementary Figure S3) has been added, to show the average microbiota composition in infant fecal samples, breast milk samples and mother’s fecal samples (lines 288-289). The new version of the Supplementary Tables & Figures has been uploaded. Numbering of the Supplementary figures has been adjusted accordingly.
  • (also the Figures 2, 4, 5, and 6 have been updated to improve the readability).
  • The diversity has been specified, referring to either alpha or beta, throughout the text.
  • Lines 339-351: Fair point, we have adjusted the text to make the conclusions more generic.
  • Multiple headings of the Results section have been adjusted to cover this fair point.
  • Line Table 1: the table, as indicated in the title, indeed describes the dietary characteristics of the five different regions. In our opinion the phrasing is correct.

Reviewer 2 Report

Comments and Suggestions for Authors

Dear Authors, thank you for such an interesting paper and very well-planned study.

Below i present some major and minor comments and suggestions that could be implemented while revising your manuscript: 

- I would recommend a slight modification of the title of the paper and using 'mother' instead of 'mum'

- Regarding the abstract, it would be beneficial if it would begin with one sentence that will act as an introduction instead of beginning it with the aim of the paper

- Line 15, abstract - which methods were used to identify the microbiota composition should be mentioned in the abstract

- Also I would recommend mentioning the investigated areas in the abstract as further in the abstract you are mentioning one and readers do not know about the rest.

- also, the mean age of the mothers should be provided in the abstract

- I would recommend using 'mother' instead of 'mum' in the whole text, it is a more scientific language

-  please check the author guidelines and adjust and correct the references and citation style according to the requirements of the journal

- in the results, demographics of the cohort, you should also add the number of participants and the mean age of participants. also, it would be beneficial to indicate how many women were from various areas indicating a number

- Line 285 - error! reference source not found - what does it mean? Please correct it as it can be found also in other parts of the text

- Line 282 - I would recommend changing the head of this paragraph. you are presenting results in this section and the title of this paragraph sounds rather like a discussion

- Table 1 - how are you 100% sure that these particular diets are commonly used in the areas chosen for this study?

- Line 547 - what about cutting this from the discussion and add as a separate 'conclusions' paragraph?

Best regards

Comments on the Quality of English Language

English should be corrected in the whole manuscript, there are some grammatical and interpunction mistakes, also some words are quite misleading and should not be included in the scientific article specifically. This should be corrected.

Author Response

We would like to thank the reviewer for the evaluation of the manuscript and the very valuable points raised.

Please find below our sincere efforts to address all.

  • The title has been rephrased, throughout the manuscript “mum” has been replaced by “mother”. Please note: adjusted title not adjusted in the “Citation” in the left margin of the front page.
  • An introductory sentence has been added to the Abstract.
  • Line 15: the methods applied to determine the microbiota composition have been included in the Abstract.
  • The investigated areas have now been addressed in the Abstract.
  • The mean age of the mothers and infants have been addressed in the Abstract, as well in the Materials & Methods section, lines 108-111.
  • Throughout the manuscript “mum” has been replaced by “mother”.
  • The references have been adjusted in line with the authors guidelines (ACS-style).
  • The number of participants, the mean age of the participants, and the number of women from the 5 distinct regions have been added to the Materials & Methods section, line 108-111.
  • Line 285: The ERROR (due to an incorrect reference to a Figure) has been resolved (the right Figure number has been included), also at other positions in the manuscript.
  • Line 282: Multiple headings of the Results section have been adjusted to cover this fair point.
  • Table 1: the description of the common diets in the five geographic regions are based on literature and the consultation of local experts. Although the diets could have some variations, the generic denominator holds.
  • Lines 547-555: On basis of this highly appreciated suggestion, we accommodated this part of the Discussion in a separate “Conclusions” paragraph.

Round 2

Reviewer 2 Report

Comments and Suggestions for Authors

Dear Authors,

Thank you very much for correcting the manuscript according to my suggestions.

I have no further comments

Kind regards

Comments on the Quality of English Language

I have no further comments